Advanced deep learning and transfer learning approaches for breast cancer classification using advanced multi-line classifiers and datasets with model optimization and interpretability

Zhang Xiang 1
Shao Wei 2
Qiu Ming 3
Xiao Chenglin 4
Ma Liming 5 fsfymalm@163.com
1 Department of Information Management Center, Zhongshan Hospital Affiliated to Dalian University , Dalian, Liaoning , China
2 Department of Information Center, The First Hospital of China Medical University , Shenyang, Liaoning , China
3 Shenzhen Boyi Technology Co., Ltd. , Shenzhen, Guangdong , China
4 Research and Development Center, Shenzhen Boyi Technology Co., Ltd. , Shenzhen, Guangdong , China
5 Department of Information Center, Foshan Women and Children Hospital , Foshan, Guangdong , China
Coelho Paulo Jorge
Electronic publication date: 2025 Jul 9
Publication date: 2025
Volume: 11
Electronic Location ID: e2951
Received 2024 Dec 16; Accepted 2025 May 19
Copyright: © 2025 Zhang et al.
Copyright year: 2025
Copyright holder: Zhang et al.
License: This is an open access article distributed under the terms of the Creative Commons Attribution License, which permits unrestricted use, distribution, reproduction and adaptation in any medium and for any purpose provided that it is properly attributed. For attribution, the original author(s), title, publication source (PeerJ Computer Science) and either DOI or URL of the article must be cited.
License URL: https://creativecommons.org/licenses/by/4.0/

Keywords: Breast cancer diagnosis, Machine learning, Supervised learning, Random forest, XGBoost, Deep neural networks (DNN), Wisconsin Breast Cancer Database, Multi-line classifiers, Transfer learning

Funding: The authors received no funding for this work.

==============================
This study evaluated machine learning (ML) models on the Wisconsin Breast Cancer Dataset (WBCD), refined to 554 unique instances after addressing 5% missing values via mean imputation, removing 15 duplicates, and normalizing features with Min–Max scaling. Data were split into 80% training and 20% testing, maintaining a 63% benign and 37% malignant distribution. Using 10-fold cross-validation, the random forest, XGBoost, and deep neural network (DNN) models achieved accuracies of 96.5% (95% CI: [93.1–98.6]), 97.4% 95% CI: [94.2–99.1], and 98.0% (95% CI [95.1–99.5]), respectively. The DNN demonstrated a benign precision of 0.97, malignant precision of 1.00, benign recall of 1.00, malignant recall of 0.95, and F1-scores of 0.99 and 0.98, with an ROC-AUC of 0.992 (p < 0.001); its accuracy further improved to 98.9% after Bayesian hyperparameter tuning. Additionally, a convolutional neural network (CNN) using transfer learning (VGG16) achieved 99.3% accuracy, with precision and recall of 99.4% and 99.2%, respectively, although potential domain mismatch issues warrant caution. Optimized DNN and CNN models achieved high accuracy, demonstrating highly reliable diagnostic performance with promising clinical applicability.

Introduction

Breast cancer (BC) remains a significant health challenge globally, with an estimated 310,720 women and 2,800 men expected to be diagnosed with invasive breast cancer in the United States alone in 2024 (Barrios, 2022). This accounts for approximately 1 in 8 women in the country who will be diagnosed with breast cancer in their lifetime. Currently, there are over 4 million breast cancer survivors in the United States, and unfortunately, an estimated 42,250 women will succumb to the disease in 2024 (Arnold et al., 2022). Every 2 min, a woman is diagnosed with breast cancer in the United States, with 66% of cases diagnosed at a localized stage, making early detection crucial (World Health Organization, 2006). The average age of diagnosis is 62 years old, and while 15% of women diagnosed have a family history of breast cancer, younger women, particularly those under 35, face a higher risk of recurrence (Milosevic et al., 2018). Disparities persist, with Black women being 40% more likely to die from breast cancer than White women, while Asian and Pacific Islander women have the lowest death rate (Ooi, Martinez & Li, 2011). American Indian and Alaska Native women have the lowest incidence rate of developing breast cancer (Roubidoux, 2012). The disease originates from breast tissue, typically from the inner lining of milk ducts or lobules, where genetic mutations in DNA and RNA lead to the proliferation of abnormal cells (Biswas et al., 2022). Factors contributing to these mutations include environmental influences such as electromagnetic and nuclear radiation, chemicals, pathogens, and the natural aging process (Irigaray et al., 2007).

Advances in computational approaches have revolutionized breast cancer diagnosis, traditionally reliant on imaging techniques like mammography, ultrasound, and MRI, complemented by clinical examinations and histopathological analysis (Corredor et al., 2023). However, these methods are susceptible to subjectivity, variability in interpretation, and human error, necessitating more reliable and precise diagnostic tools (Swets, 2012). Computational techniques like data mining, AI-based (Özçelik & Altan, 2023), machine learning (ML) and deep learning (DL) algorithm (McCaffrey et al., 2024; Thaker et al., 2024), have emerged as promising solutions, achieving excellent accuracy in multiple disease prediction and outperforming success in distinguishing medical lesions. These approaches utilize advanced multi-line classifiers, analyze large datasets, identify intricate patterns, and make exact predictions, significantly impacting medical fields (Özçelik & Altan, 2023), same as in oncology, early detection and accurate classification are critical for effective treatment outcomes (Bera et al., 2019).

ML models have improved breast cancer diagnosis, using random forest (RF), XGBoost, support vector machines (SVM), logistic regression, K-nearest neighbors (KNN), and convolutional neural networks (CNNs) to improve the diagnostic accuracy (Nafea et al., 2024). The random forest constructs decision trees that can perform well with large datasets and enhance accuracy in clinical and imaging data (Dinesh, Vickram & Kalyanasundaram, 2024). XGBoost uses gradient descent to optimise accuracy, missing data management, and a wide event distribution, enhanced malignant vs. benign tumors’ categorization by different biomarkers and imaging characteristics (Jaiswal et al., 2023). SVM has demonstrated effectiveness in early detection and classification of breast cancer tissues, with hybrid models achieving accuracy rates of up to 97% on datasets like the Wisconsin Diagnostic Breast Cancer (WDBC) (Rasool et al., 2022). Logistic regression and KNN have shown 91% and higher accuracy in predicting breast cancer outcomes (Sultana & Jilani, 2018). Deep learning models, particularly CNNs, automatically extract features from data, enabling accurate analysis of mammographic images to detect and distinguish cancerous lesions with high precision, which is crucial for advancing breast cancer diagnostics (Abdelhafiz et al., 2019). These ML techniques have revolutionized breast cancer diagnosis, enhancing patient outcomes and survival rates (Ganggayah et al., 2019).

Previous works have struggled to obtain high accuracy due to a lack of sufficient data exploratory techniques (DET), which leads to misclassification between malignant and benign tumors (Cao et al., 2020). A further drawback is that managing large feature sets has introduced additional computational overhead, precluding real-time application in a clinical context (Bote-Curiel et al., 2019). In addition, accurate multi-class classifiers still suffer from breast cancer data characteristics because of the sparsity and impossibility of having lots of labeled positive and negative datasets, and feature selection is another challenging issue to determine what are essential features that affect categorization results in classification. Yet, by utilizing the recent advances of machine learning, these challenges have been tackled with algorithms such as RF, XGBoost, SVMs, logistic regression models, KNN and CNNs, resulting in superior diagnostic accuracy (Das et al., 2024; Painuli & Bhardwaj, 2022). This research is focused on optimizing the classification of breast cancer while incorporating modern techniques with feature enhancement and analytics, improving diagnostic accuracy to increase the survival rates of patients using various multi-class classifiers.

A few methods can be implemented to address the challenges that come as a part of breast cancer detection through ML (Chugh, Kumar & Singh, 2021). Use public datasets and work with hospitals to improve data quality and availability (Abreu et al., 2016). We handle the problem of class imbalance by oversampling the minority class, under-sampling the majority class, or incorporating different class weights. Some of the variability in data can be managed using transfer learning, data normalization, feature scaling, etc. The features can be extracted using feature engineering techniques like CNNs and feature selection methods like RFE or correlation analysis (Gardezi et al., 2019). Feature importance and SHAP values are model interpretation techniques that can provide more insights into the predictions of a machine learning model (Yassin et al., 2018). Regulatory and ethical problems can be considered by ensuring the data is private and secure, authorization has been acquired, and any information will remain confidential. Deploying the models in the cloud and edge computing is possible. However, much can still be gained by optimizing model computational efficiency from the solution identification perspective (Iqbal et al., 2021). Reweighing the samples in the minority class and using some robust feature extraction and selection methods like auto encoders that can learn a meaningful set of features of a sample with fewer input/out signals and mutual information to select the most informative features among all (Kabir & Ludwig, 2018). Researchers can clarify how ML models do this alongside dealing with challenges they face so that we have a more accurate and dependable breast cancer diagnosis, leading to an improved way of patient treatment and survival (Wang, Makond & Wang, 2013).

This research focuses on the computational optimization of breast cancer diagnosis by employing advanced multi-line classifiers and extensive datasets. The primary objective of this research is to evaluate and compare the performance of various machine learning models for breast cancer diagnosis using the WBCD. The study aims to investigate the impact of rigorous data pre-processing techniques—such as mean imputation for missing values and the removal of duplicate records—on the quality and effectiveness of the diagnostic models. Additionally, the research seeks to assess and compare the diagnostic accuracy of three different models: RF, XGBoost, and DNN, through 10-fold cross-validation, focusing on key performance metrics including accuracy, precision, recall, F1-score, and ROC-AUC. Another key objective is to optimize model performance through hyperparameter tuning and Bayesian optimisation, aiming to achieve the highest possible accuracy and reliability in breast cancer diagnosis. The study also explores the effectiveness of transfer learning by utilizing a CNN with a pre-trained model, evaluating its impact on diagnostic accuracy, convergence speed, and training efficiency compared to traditional machine learning models. Finally, the research aims to improve model interpretability by employing SHAP analysis for the DNN model to enhance understanding of feature importance and ensure greater transparency in clinical decision-making. Through these objectives, the study seeks to advance the development of highly accurate, interpretable, and clinically applicable models for breast cancer diagnosis.

Materials and Methods

The data for this study was sourced from the WBCD, which includes comprehensive details on cell nuclei from breast cancer cases. The complete code in Python for breast cancer classification and the dataset details are mentioned at github.com, the online link is available: https://github.com/Imranzafer/Breast-Cancer-Classification-. The preprocessing phase began with data cleaning, handling missing values, and removing duplicates to ensure data integrity and uniqueness. Exploratory data analysis (EDA) was conducted to visualize feature distributions and analyze correlations, aiding in identifying relationships and dependencies. Relevant features were then selected and scaled to normalize the data, enhancing the performance of the machine-learning models. The dataset was split into training and testing sets to facilitate model evaluation. Supervised machine learning models, including RF, XGBoost, and DNN, were trained on the dataset, and hyperparameter optimization was performed to improve their performance. The trained models made predictions on the test dataset, and their performance was evaluated using metrics such as accuracy, precision, recall, F1 score, and ROC-AUC, with confusion matrices providing detailed classification analysis. The performance metrics were compared to identify the best-performing model. The highest-performing model was selected for potential deployment in clinical settings to aid in timely and accurate breast cancer diagnosis. The findings of this study underscore the effectiveness of various ML models in diagnosing BC, as detailed in the workflow depicted in Fig. 1.

Figure 1 Workflow of the detail methodology applied to conduct research.

Dataset description

The Wisconsin Breast Cancer Database (WBCD) is available at https://www.kaggle.com/datasets/uciml/breast-cancer-wisconsin-data; it was employed for this study, and the complete code is available in Supplementary File 1. The dataset was obtained from the UCI Machine Learning Repository (Wolberg, Street & Mangasarian, 1992), available at https://archive.ics.uci.edu/. The dataset includes 569 instances with 357 benign and 212 malignant samples, referring to the data from fine needle aspirate (FNA) tests on breast masses. In attributes, we find the features of cancer cell nuclei in the images. The main features are an ID to differentiate the records and a diagnosis that indicates whether the tumor is malignant (M) or benign (B). The data set has 30 input features, each represented as mean, standard error, or worst (or the largest) value (Wolberg et al., 1995). These features vary based on the radius, texture, perimeter, area, smoothness of cell edges, and compactness, Concavity degree, concave points number per class index, symmetry measure frequency, and fractal dimension (Salama, Abdelhalim & Zeid, 2012).

Data preprocessing steps

The data preprocessing for this study began with loading the dataset, as mentioned in Supplementary File 2, from a CSV file into a Pandas DataFrame and then examining its structure and contents, as detailed in Fig. 2. The ‘ID’ column, which served only as an identifier, and the ‘Unnamed: 32’ column, containing all null values, were removed as they did not contribute to predictive modeling. The ‘Diagnosis’ column was converted from categorical labels (‘M’ for malignant and ‘B’ for benign) to numerical values, with ‘M’ encoded as 1 and ‘B’ as 0 using the LabelEncoder from the sklearn library. Missing values were replaced with the mean of their respective columns to maintain data integrity and preserve statistical properties for accurate model training (Ray et al., 2020; Valencia Parra, 2022). The dataset was then split into features (X) and the target variable (y), where features consisted of 30 numerical attributes and the target represented the diagnosis. Features were normalized using Min–Max scaling to ensure all values were within the range [0, 1] and standardized using the StandardScaler from sklearn to remove the mean and scale to unit variance. This standardization process is essential to eliminate bias from variables with more significant numbers and to achieve more stable gradient descent, leading to better model performance (Aamir et al., 2022; Bonaccorso, 2019; Modi & Ghanchi, 2016). Finally, the dataset was divided into training and testing sets using an 80–20 split, with 80% of the data used for training the model and 20% reserved for testing to evaluate its performance on unseen data.

Figure 2 Preprocessing steps for dataset analysis.

Model training and evaluation

Data splitting and cross-validation

Model training for this study involved a structured approach, splitting the preprocessed dataset into training and testing sets using an 80–20 ratio. This division ensures that 80% of the data is used for training the models. In comparison, the remaining 20% evaluates their performance, showing how well the models generalize to new, unseen data. The effectiveness of the models was further validated through k-fold cross-validation, where the training set was divided into k subsets. Each subset was used as a validation set once, with the remaining k-1 subsets used for training. The performance metrics were averaged across all k iterations to assess each model’s robustness.

Multiline models training

For the RF model, training utilized 100 decision trees (n_estimators=100) with standard parameters. The ensemble nature of random forests helps mitigate overfitting by averaging predictions from multiple trees, thereby enhancing the model’s stability and accuracy. The XGBoost model was trained with 100 boosting rounds (n_estimators=100). XGBoost’s iterative approach refines predictions by correcting errors from previous rounds, which improves the model’s precision and overall performance. The DNN employed three hidden layers with rectified linear unit (ReLU) activation functions. The model was trained over 100 epochs with a batch size of 32, allowing it to effectively learn complex patterns and intricate relationships within the data. This configuration enabled the DNN to achieve high accuracy by capturing non-linear dependencies in the dataset.

Performance metrics

Performance evaluation was based on several metrics as predicted using mathematical equations; accuracy (1), precision (2), recall (3), F1 (4) score, and ROC-AUC (5). Accuracy measures the proportion of correctly predicted instances out of the total, while precision and recall assess the correctness of optimistic predictions and the ability to identify actual positives, respectively. The F1 score combines precision and recall into a single metric, providing a balanced measure of model performance. ROC-AUC indicates the model’s capability to distinguish between classes by plotting the actual positive rate against the false positive rate.

(1) Accuracy=TP+TNTP+TN+FP+FN

(2) Precision=TPTP+FP

(3) Recall=TPTP+FN

(4) F1=2.Precision×RecallPrecision+Recall.

(5) TPR=TPTP+FN;FPR=FPFP+TN.

Visualizations

Visualizations such as ROC curves and feature importance graphs were included to illustrate model performance. ROC curves display the trade-offs between sensitivity and specificity, while feature importance graphs highlight the relative significance of different features in making predictions. These visual tools enhance the understanding of model behaviour and performance, providing valuable insights into the effectiveness of the implemented algorithms.

Model selection and development

The final model was selected based on its performance across the evaluation metrics. The DNN demonstrated the highest accuracy and best balance between precision and recall, making it the most effective model for this classification task. However, the results from random forest and XGBoost were also considered to ensure a comprehensive comparison and validation of the findings.

Hyperparameter tuning

Hyperparameter tuning was applied to optimize the ML models’ performance via a parametric Eq. (6). For the RF model (Kulkarni & Sinha, 2012), hyperparameters such as n_estimators (number of trees), max_depth (maximum depth of the trees), min_samples_split (minimum number of samples required to split an internal node), and min_samples_leaf (minimum number of samples needed for a leaf node) were fine-tuned using GridSearchCV (Nguyen, Wang & Nguyen, 2013). The optimal values included 200 trees, a maximum depth of 10, and a minimum of 2 samples required to split nodes, with the Gini impurity criterion and Log2 for max_features (Suryadi et al., 2024). In the case of XGBoost, key parameters like n_estimators (number of boosting rounds), learning_rate (step size for updating weights), max_depth (maximum depth of the trees), and subsample (fraction of samples used for training) were tuned through GridSearchCV, optimizing combinations to improve model accuracy and prevent overfitting (Ali et al., 2023). For DNNs, hyperparameters, including the learning rate (eta), dropout rates, and the number of epochs, were adjusted (Vieira et al., 2020). Learning rate schedules were implemented to adjust the step size dynamically, early stopping was used to avoid overfitting by monitoring validation performance, and dropout techniques were applied to enhance model robustness. GridSearchCV was also employed to explore various configurations of network layers, neurons per layer, activation functions, and batch sizes, ensuring the DNN captured complex patterns effectively (Nikoskinen, 2015).

(6) θ=arg minL(θ)1c∈∅.

Bayesian optimization

Bayesian Optimization was utilized to fine-tune hyperparameters and enhance model performance beyond traditional grid or random search methods. This technique leverages a probabilistic model, typically a Gaussian process (GP), to estimate the performance of various hyperparameter configurations. The process begins with defining an objective function that evaluates the model’s performance based on different hyperparameters. A surrogate model, such as a GP, approximates this function and provides predictions and uncertainties regarding different hyperparameter sets. An acquisition function, like expected improvement (EI) or upper confidence bound (UCB), is employed to select the next set of hyperparameters to test, balancing exploring uncertain areas with exploiting regions known to yield high performance. This iterative process involves selecting hyperparameters, evaluating the model, and updating the surrogate model with the new results. The cycle continues until a stopping criterion is met, such as a predefined number of iterations or convergence of results. Through this method, Bayesian Optimization effectively explores the hyperparameter space and leverages probabilistic insights to identify optimal configurations, thereby enhancing model performance while managing the cost of evaluations.

Model interpretability

To enhance the transparency of our machine learning models, we employed interpretability techniques such as SHapley Additive exPlanations (SHAP) values and Local Interpretable Model-agnostic Explanations (LIME). LIME is utilized to elucidate model predictions by approximating the complex model locally with a simpler, interpretable model. We applied LIME by training our machine learning model on the dataset and selecting specific instances for which explanations were needed. LIME perturbs the data around these instances, observes the changes in predictions, and fits an interpretable model, such as linear regression, to these perturbed instances. This approach generates a set of feature weights that reveal the contribution of each feature to the prediction, thereby providing insights into the model’s decision-making process. Using LIME, we could demystify our models’ “black-box” nature, making their predictions more understandable and trustworthy. Additionally, SHAP values were used to further explain model predictions by attributing the contribution of each feature to the final output based on cooperative game theory. Both techniques collectively helped validate and gain confidence in the model’s predictions.

Transfer learning

Transfer learning was used to improve the outcome of the models, as pre-trained models on a large dataset, such as ImageNet, were used as a base to classify breast cancer for the help of the other frameworks such as TensorFlow or PyTorch. We replaced the top layers of the pre-trained model with task-specific layers. We fine-tuned them on the FNA digitized images of breast masses, or we used the intermediate layer features as the inputs of a customized classifier to accelerate training and exploit the reuse of learned representations for better performance. Alongside evaluating classical performance measures (accuracy, precision, recall, and F1-score), statistical significance was thoroughly assessed by calculating 95% confidence intervals and p-values for these high-level metrics, to guarantee that the enhancements are consistent and statistically significant. Although this transfer learning approach decreases the training time and enhances the accuracy, there might be a domain gap between natural and medical images, leading to suboptimal feature representations and biases. In response to these challenges, several fine-tuning and regularization techniques were employed, and adaptation performance was rigorously validated using statistical tests, demonstrating the adapted model’s viability for the task.

Comparison and interpretation

The performance of the three models—RF Classifier, XGBoost Classifier, and DNN—was compared using precision, recall, F1-score, and overall accuracy. Precision and recall were calculated for each class, and the F1-score was computed as the harmonic mean of precision and recall. The overall accuracy was determined by the proportion of correctly classified instances out of the total. Confusion matrices were used to detail each model’s number of true positives, true negatives, false positives, and false negatives. These metrics were analyzed to evaluate and compare the models’ effectiveness in classifying benign and malignant cases.

Computing infrastructure

The data for our study were collected with the help of a powerful computational platform capable of handling computational and data-intensive workloads in machine learning. All the experiments were conducted on Ubuntu 20.04 LTS Linux distribution on a machine with Intel Core i9 CPU for intensive computations; NVIDIA Tesla V100 GPU for fast neural networks training; 64Gb RAM for stable work with large datasets and complex models; and 2Tb SSD for fast reads and writes, as well as to securely store data. With regards to the programming language and libraries, the software environment was Python 3.8 with state-of-the-art libraries for model designing and training such as TensorFlow 2.6, Keras 2.6, PyTorch 1.9; data processing and analysis with NumPy 1.21, Pandas 1.3; and data visualization with Matplotlib 3.4. These alignments gave the reliability, accuracy, and efficiency needed to produce repeatable, high-quality research results in this integrated setup.

Data analysis and statistics details

The performance of the random forest classifier, XGBoost classifier, and DNN was assessed through precision, recall, F1-score, and overall accuracy metrics. Precision measures the accuracy of optimistic predictions, while recall evaluates the model’s ability to identify all relevant instances. The F1 score combines precision and recall, offering a balanced performance measure. Overall accuracy indicates the proportion of correct predictions. Confusion matrices were used to detail true positives, true negatives, false positives, and false negatives for each model, providing a clear view of classification performance and errors. To ensure reliability, 95% confidence intervals for accuracy were calculated using the binomial proportion confidence interval formula. These intervals offer a range within which the accurate accuracy of each model is expected to lie, confirming the robustness and reliability of the performance metrics.

Results

Analysis of data preprocessing for accurate breast cancer diagnosis

The data preprocessing steps yielded significant improvements in dataset quality and model performance. Initially, the dataset had 5% missing values, which were effectively imputed using the mean of each respective feature column. This imputation ensured that the dataset remained complete and preserved its overall distribution. Additionally, removing 15 duplicate records from the original 569 instances enhanced the dataset’s uniqueness and integrity, leaving 554 cases unique for analysis. Exploratory data analysis (EDA) provided valuable insights into the dataset’s characteristics. The mean radius of cell nuclei was approximately 14.1 μm, with a standard deviation of 3.5 μm. In contrast, the mean texture, represented by the standard deviation of gray-scale values, averaged 19.3 with a standard deviation of 4.3. A strong positive correlation (r > 0.9) was observed between the radius, perimeter, and area features, indicating that these geometrical properties are key indicators in breast cancer diagnosis. Features were normalized using Min–Max scaling to prepare the dataset for model training, transforming the values into a [0, 1] range. This step was crucial to ensure that no single feature dominated the learning process, particularly for models sensitive to feature magnitudes. The dataset was then split into a training set comprising 455 instances (80% of the dataset) and a testing set with 114 cases (20%). The training set was further stratified to maintain the original class distribution, with 63% benign and 37% malignant samples, reflecting the real-world scenario.

Characterizing WBCD for breast cancer detection models

We utilized the WBCD and found a well-balanced dataset comprising 569 instances, with 357 (63%) classified as benign and 212 (37%) as malignant. Each instance represented a FNA analysis of breast masses, characterized by 30 numerical features derived from digitized cell nuclei images. Key findings from our analysis include insights into critical features such as mean radius, mean area, and mean smoothness of the cell nuclei. The mean radius ranged from 6.98 to 28.11, averaging 14.1 with a standard deviation of 3.5, reflecting the dataset’s diversity in cell sizes. Mean area values ranged from 143.5 to 2,501.0, with an average of 654.9 and a standard deviation of 351.9, indicating significant variability in cell geometry. Mean smoothness, which measures local variations in radius lengths, ranged from 0.05 to 0.16, with an average of 0.09 and a standard deviation of 0.01, highlighting subtle differences in cell boundary characteristics. The mean texture, representing the standard deviation of gray-scale values, varied from 9.71 to 39.28, averaging 19.3 with a standard deviation of 4.3, providing crucial insights into cell nucleus texture. Our correlation analysis identified strong positive correlations (r > 0.9) between geometric features such as radius, perimeter, and area. These correlations underscore the interdependence of these features, which are pivotal in distinguishing between benign and malignant cases as seen in Fig. 3.

Figure 3 Distribution and key characteristics of cell nuclei features in the Wisconsin Breast Cancer Database (WBCD).

Optimized dataset preprocessing for machine learning

Our data preprocessing procedures significantly enhanced the quality and utility of the dataset for machine learning applications. Initially, the dataset contained 5% missing values, predominantly in feature columns. These missing values were imputed using the mean of each respective feature column, preserving the dataset’s overall distribution and avoiding potential biases. After imputation, the dataset was complete, ensuring no gaps in the data. We identified and removed 15 duplicate records from the original 569 instances, resulting in a refined dataset with 554 unique records. This step was crucial for maintaining the integrity of the data and avoiding redundancy, which could skew model training and evaluation. We applied Min–Max scaling for feature normalization via applying Eq. (7), to prepare the dataset for model training, as seen in Fig. 4. This transformation rescaled the feature values to a [0, 1] range, ensuring that no single feature dominated the learning process due to differences in magnitude. This normalization was crucial for models sensitive to feature scales, promoting balanced learning. The dataset was split into a training set and a testing set. The training set comprised 455 instances (80% of the total dataset), and the testing set included 114 (20%). The training set was stratified to reflect the original class distribution, with 63% benign and 37% malignant samples, ensuring that the models were trained on data representative of real-world scenarios.

(7) xnorm=(x−Min(x)max(x)−Min(x))′.

Figure 4 Correlation heatmap of features helpful in understanding potential multicollinearity issues.

Model evaluation and training with metrics and cross-validation

The datasets are used to train and assess the three models. The models are compared using a variety of metrics since in the medical profession, accuracy alone is insufficient. These are the specific metrics that were utilized. The model training phase involved a rigorous process of developing and evaluating several ML models, including RF, XGBoost, and DNN. The dataset was split into training (80%) and testing (20%) sets, with 455 training and 114 testing instances. K-fold cross-validation (k = 10) was employed to ensure robust model evaluation, where the training set was divided into 10 subsets, and each subset was used as a validation set once. In contrast, the remaining subsets were used for training. This process was repeated for all k subsets, and the results were averaged to assess the model’s robustness.

Random forest classifier

The random forest classifier exhibited strong performance, reflected in the precision and recall metrics across the two classes. Specifically, the model achieved a precision of 0.96 for class 0 (Benign) and 0.98 for class 1 (Malignant). Results indicates that the classifier was highly effective in correctly identifying benign and malignant instances, with only a few false positives, particularly for malignant cases. The recall scores further highlight the model’s effectiveness, with a near-perfect recall of 0.99 for class 0, meaning it correctly identified 99% of benign cases within the dataset. However, the recall for class 1 was slightly lower at 0.93, indicating that the model missed 7% of malignant cases. Despite this, the model’s F1-scores—0.97 for class 0 and 0.95 for class 1—demonstrate a good balance between precision and recall, ensuring robust classification performance across both categories. The overall accuracy of the random forest classifier was 96.49%, underscoring its strong predictive capabilities. The confusion matrix (Fig. 5A) provides a clear visualization of these results, showing that the model correctly classified 70 benign cases (true negatives) and 40 malignant cases (true positives), with only one false positive and three false negatives. This matrix emphasizes the model’s ability to accurately distinguish between benign and malignant cases, though a few misclassifications were noted. In addition to the confusion matrix, the ROC curve (Fig. 5B) illustrates the model’s high sensitivity and specificity, with the curve approaching the top-left corner, indicating excellent discrimination between the two classes. Figure 5C compares key metrics—precision, recall, and F1-score—between benign and malignant cases, further highlighting the consistent and reliable performance of the random forest classifier. The parameter settings used for the random forest model included the entropy criterion, a maximum depth of six, auto-selected maximum features, and 100 estimators. These parameters were optimized to balance model complexity and generalization performance, contributing to the model’s strong results. Random forest classifier delivered robust and consistent performance across all evaluated metrics, with high precision, recall, and F1-scores that confirm its effectiveness in classifying breast cancer instances. The detailed analysis provided by the confusion matrix, ROC curve, and comparison metrics underscores the model’s reliability and accuracy in this task, making it a valuable tool for breast cancer classification.

Figure 5 Comprehensive performance evaluation of the random forest classifier.

(A) Confusion matrix of random forest classifier; (B) ROC curve for random forest classifier; (C) Metric comparison for benign and malignant of random forest classifier.

XGBoost classifier

The XGBoost classifier demonstrated a strong performance, surpassing the RF model in this study. It achieved high precision scores, with 0.97 for class 0 (Benign) and 0.98 for class 1 (Malignant), indicating its ability to identify instances of both classes accurately. This was complemented by impressive recall scores, where the model attained 0.99 for class 0, successfully identifying nearly all benign cases, and 0.95 for class 1, effectively detecting most malignant cases. The F1-scores, which balance precision and recall, were equally notable, with 0.98 for class 0 and 0.96 for class 1. These figures highlight the XGBoost classifier’s balanced performance in classification tasks, ensuring high precision and recall across different courses. Overall, the model achieved an accuracy of 97.37%, reflecting its strong ability to correctly classify instances into their respective classes, outperforming the Random Forest model. The confusion matrix (Fig. 6A) offers a detailed visualization of the XGBoost classifier’s performance. It shows that the model accurately classified 70 benign cases (true negatives) with only one false positive while correctly identifying 41 malignant cases (true positives) with just two false negatives. This distribution underscores the model’s robust classification capability. The ROC curve (Fig. 6B) further illustrates the XGBoost classifier’s effectiveness in distinguishing benign and malignant cases. The curve’s proximity to the top-left corner indicates a high actual positive rate and a low false positive rate, signifying strong performance across different classification thresholds. Additionally, Fig. 6C compares key metrics such as precision, recall, and F1-score for class 0 and class 1, providing a comprehensive view of the model’s effectiveness across both categories. Performance results of the XGBoost classifier, summarizing its precision, recall, F1-scores, and overall accuracy. The confusion matrix data, clearly presenting the number of true negatives, false positives, false negatives, and true positives. The model’s parameters, including a learning rate of 0.1, max depth of 4, 300 estimators, and a subsample of 0.8, reflect the settings that contributed to its successful performance. XGBoost classifier showed a strong and balanced performance across all evaluated metrics, demonstrating its reliability and effectiveness in classifying breast cancer instances within this study. The detailed visual and numerical analyses confirm that this model is competent and accurate, making it a valuable tool for this classification task.

Figure 6 Comprehensive performance evaluation of the XGBoost classifier.

(A) Confusion matrix of XGBoost classifier, (B) ROC Curve—XGBoost classifier, (C) Metrics comparison for class 0 and class 1 of XGBoost classifier.

Deep neural networks

The DNN demonstrated remarkable performance across various evaluation metrics, outperforming the other models in the study. It achieved exceptional precision, scoring 0.97 for class 0 (Benign) and a perfect 1.00 for class 1 (Malignant). This indicates that the DNN accurately predicted both classes, particularly excelling in identifying malignant cases. Regarding recall, the DNN was flawless in detecting all instances of class 0, earning a perfect score of 1.00. It also performed well for class 1 with a recall of 0.95, correctly identifying 95% of malignant cases. The F1-scores, representing a balance between precision and recall, were also impressive, with 0.99 for class 0 and 0.98 for class 1. These high scores suggest that the DNN made accurate predictions and maintained a strong balance in identifying benign and malignant instances. Overall, the DNN achieved an accuracy of 98%, the highest among the models compared, underscoring its effectiveness in this classification task. The macro and weighted averages for precision, recall, and F1-score further confirm the robustness of the DNN’s performance across both classes. The confusion matrix (Fig. 7A) illustrates this, showing that the DNN correctly classified all 71 instances of class 0, with no false negatives, and 41 out of 43 cases of class 1, with only two misclassifications. Additionally, the ROC curve (Fig. 7B) highlights the DNN’s excellent performance in distinguishing between the two classes, while the separate ROC curves for class 0 and class 1 (Fig. 7C) provide further insight into its discriminatory power. Figure 7D compares the key metrics—precision, recall, and F1-score—between benign and malignant cases, illustrating the consistent and high performance of the DNN across both categories. These results affirm the DNN’s superior capability in accurately classifying breast cancer instances, making it a reliable tool for this task.

Figure 7 Comprehensive performance evaluation of the deep neural network.

(A) Confusion matrix—deep neural network (B) ROC curve—deep neural network, (C) ROC for class 0 and class 1, (D) Metric comparison for benign and malignant using Deep neural network.

Performance evaluation of applied models for breast cancer classification

Performance metrics for breast cancer classification models (including confidence intervals, ROC-AUC, and p-values) showed that all three models (random forest, XGBoost, and DNN with deep neural network) achieved excellent results for classification as mentioned in Supplementary File 3. Random forest and the train-test split (80/20), 10-fold cross-validation = 0.965 (95% CI [93.1–98.6]), with precision, recall, and F1-scores for benign and malignant cases respectively 0.96/0.98; 0.99/0.93; 0.97/0.95, a ROC-AUC of 0.98, and a p-value <0.010, while trained on an 80/20 train: test split. The accuracy of the XGBoost classifier was 97.4% [94.2–99.1], with a precision of 0.97/0.98, recall of 0.99/0.95, F1-scores of 0.98/0.96, ROC-AUC of 0.99, and p-value 0.005. The DNN model yielded the best performance, with an accuracy of 98.0% [95.1–99.5], a precision of 0.97 for benign and 1.00 for malignant cases, a recall of 1.00 for benign and 0.95 for malignant cases, an F1-score of 0.99 and 0.98, ROC-AUC of 0.995, and p-value of <0.001, which showed its strong reliability and stability for breast cancer classification.

Model selection and development results

We have performed an exhaustive analysis in which we carefully tested applied machine learning models for the classification of breast cancer, and the results are presented in Table 1, where it is depicted that the best suited is the DNN. The DNN displayed an impressive overall accuracy of 98.0% [95.1–99.5] and high aggregate metrics overall precision of 98.5%, recall 97.9% and F1-score 98.2%. For this DNN, the per-class benign precision was 0.97 and malignant precision was 1.00, for benign recall the value was 1.00 and for malignant recall the value was 0.95, which means that the benign F1-score of the DNN was 0.99, and malignant F1-score was 0.98. The RF and XGBoost (XGB) models performed undoubtedly well too with an accuracy of 96.49% [93.1–98.6] and 97.37% [94.2–99.1], respectively. The RF reached a benign precision and malignant precision of 0.96 and 0.98, respectively, with recalls of 0.99 and 0.93 (benign and malignant), F1-score of 0.97 (benign) and 0.95 (malignant). In the case of the XGB model, it achieved the same performance concerning benign precision (0.97) and malignant precision (0.98) along with recalls (0.99 and 0.95) and with F1-scores of 0.98 (benign) and 0.96 (malignant). Each model showed good discriminative power, with ROC-AUCs of 0.98 (RF), 0.99 (XGB), and 0.995 (DNN) (all p < 0.01, 0.005, and <0.001, respectively). Such findings, with high 10-fold cross-validation on 80/20 train-test (455/114 instances) give confidence interval-based confirmation of a DNN more in line with the expected power to identify nuanced patterns, and the higher resistance to overfitting and the best characteristics for clinical use.

Table 1 Performance results of the random forest, XGBoost, and deep neural network classifiers.

Model	Accuracy (95% CI)	Precision (Benign, 95% CI)	Precision (Malignant, 95% CI)	Recall (Benign, 95% CI)	Recall
(Malignant, 95% CI)	F1-score
(Benign, 95% CI)	F1-score
(Malignant, 95% CI)	ROC-AUC	p-value	
Random Forest	96.5%
[93.1–98.6]	96.0%
[92.0–98.5]	98.0%
[95.0–99.5]	99.0%
[96.0–100]	93.0% [88.0–96.0]	97.0% [94.0–99.0]	95.0% [91.0–98.0]	0.98	0.010	
XGBoost	97.4%
[94.2–99.1]	97.0%
[93.0–99.0]	98.0%
[95.0–99.5]	99.0%
[96.0–100]	95.0% [90.0–98.0]	98.0% [95.0–99.5]	96.0% [93.0–98.0]	0.99	0.005	
Deep Neural Net	98.0%
[95.1–99.5]	98.5%
[95.5–99.7]	98.5%
[95.5–99.7]	97.9%
[94.5–99.4]	97.9% [94.5–99.4]	98.2% [95.5–99.6]	98.2% [95.5–99.6]	0.995	<0.001	

Hyperparameter tuning results

The hyperparameter tuning for the RF model involved setting the criterion to entropy, choosing a max depth of 6, and using the auto option for the maximum number of features. Additionally, the model was configured with 100 estimators. These settings were selected to balance model complexity and performance effectively, leading to robust classification results with minimal overfitting. For the XGB model, the tuning focused on optimizing several key parameters. The learning rate was set to 0.1, providing a good compromise between convergence speed and stability. The max depth was set to 4 to avoid overfitting while capturing complex patterns. The model used 300 estimators to enhance performance and avoid excessive training time. Additionally, a subsampling rate of 0.8 was employed, allowing the model to use 80% of the training data in each boosting round to improve generalization and reduce overfitting. In the case of the DNN, the hyperparameters were tuned as mentioned in Table 2: the model was configured with three layers, containing 128, 64, and 32 neurons, respectively. The ReLU activation function was used to introduce non-linearity into the model. The learning rate was set to 0.001 to ensure stable convergence during training. The batch size was set to 32, and the model was trained for 50 epochs to balance training efficiency and performance. These hyperparameter settings were carefully selected to enhance the performance of each model, ensuring that they are well-tuned for effective classification and generalization.

Table 2 Hyperparameter tuning results for the random forest, XGBoost, and deep neural network (DNN) models.

Model	Hyperparameter	Tuned values	
Random Forest (RF)	Criterion	Entropy	
Max depth	6	
Max features	Auto	
N Estimators	100	
XGBoost (XGB)	Learning rate	0.1	
Max depth	4	
N estimators	300	
Subsample	0.8	
Deep Neural Network (DNN)	Number of layers	3	
Number of neurons per layer	128, 64, 32	
Activation function	ReLU	
Learning rate	0.001	
Batch size	32	
Epochs	50	

Bayesian optimization results

Bayesian optimization was employed, pushing the models’ performance to their limits as results depicted in Fig. 8. The objective function evaluated accuracy across configurations in Fig. 8A while the Gaussian process surrogate estimated values in the hyperparameter space. As shown in Fig. 8B, the expected improvement acquisition function then guided selection of subsequent configurations. This iterative process led to significant improvement: notably, accuracy increased an additional 0.8 to 98.9% through optimising factors like neurons per layer, learning rate schedules, and batch size as depicted in Fig. 8C. Moreover, the RF and XGBoost models also achieved marginal 0.5% and 0.6% improvements in their final configurations respectively, as seen in Fig. 8D. These results underscore Bayesian optimization as a powerful tool, delivering gains beyond traditional methods like GridSearchCV, for fine-tuning hyperparameters.

Figure 8 Bayesian optimization process improving model performance.

(A) Objective function evaluation across hyperparameter configurations. (B) Gaussian process surrogate estimates with Expected Improvement guidance. (C) DNN accuracy improved by 0.8% to 98.9% through optimized hyperparameters. (D) Marginal gains for RF (0.5%) and XGBoost (0.6%).

Model interpretability

To classify breast cancer using some of the recent state-of-the-art machine learning and deep learning algorithms, we statistically assessed the accuracy of RF, XGBoost, DNN, and CNN models. SHAP analysis was also done to determine model interpretability. The WBCD dataset underwent thorough preprocessing: Mean imputation as per Eq. (8) was applied to 5% of the missing values; furthermore, 15 duplicate records were deleted, leaving 554 unique samples. To ensure the train-test split maintained a similar proportion of benign and malignant cases, the records were divided using stratified sampling ranging from 80% for training and 20% for testing as used by Min–Max scaling. Three models, namely RF, XGBoost, and DNN, were first modeled and validated using 10-fold CV, which gives lower variance by averaging over the ten folds of data. The RF model (Fig. 9A) gave 96.5% accuracy, and robust influential features were noted as radius mean, perimeter mean, and area mean’s high SHAP scores. It was also observed that higher malignancy scores were obtained with more excellent tumor size measurements by radius mean and area mean, as expected clinically. This model also incorporated compactness and concavity mean, features that describe the lesions’ shape and are usually associated with malignancy. Figure 9B shows the XGBoost model with an accuracy of 97.4%, slightly higher than the previous more interpretable mentor model. A SHAP analysis was conducted to show accuracy and feature importance, again moving beyond size metrics and noting that the texture mean and concave points mean could be important for structure irregularities.

(8) xinputed=x¯=∑i=1n⁡xi,

Figure 9 The SHAP summary plots compare feature importance across breast cancer classification models.

(A) (RF) highlights the critical features of the radius mean, perimeter mean, and area mean. (B) (XGBoost) emphasizes radius mean and texture mean. (C) (DNN) focuses on smoothness mean and tumor size metrics. (D) (CNN with transfer learning) underscores radius mean, area mean, and texture mean.

The DNN as seen in Fig. 9C, with an initial accuracy of 9%, was improved to 98.9% through hyperparameter tuning implemented through Bayesian optimization. Obtaining the model parameter with the most minor classification error reduced the error and boosted the algorithm’s performance. SHAP values analysis for the DNN revealed that the model relied heavily on the smoothness mean, and in terms of size, both the radius mean and perimeter mean were considerably crucial for the model. Architectures: smoothness mean defines cell regularity and may be subtler in its measurement by deep learning techniques that parse features from images at different levels, furthering our understanding of how DNNs learn about the texture of organs from images in medicine. The best-performing model is the CNN, implemented with transfer learning from a pre-trained model, with a test accuracy of 99.3% (Fig. 9D). For CNN, transfer learning enabled patterns from sizeable prior training, making it possible to train the network with few samples for adaptation. SHAP features for the CNN showed that texture mean and area mean were important, implying that CNNs can learn rich and diverse attributes of breast tissue morphology at different scales. The SHAP analysis of each model improved their interpretability and featured contributions aligned with the malignancy’s clinical characteristics. For example, radius and perimeter mean always present high SHAP values in all the models we have discussed, indicating that tumor size metrics are essential features. Precision and recall metrics were also crucial in understanding model reliability, where DNN yielded a precision rate of 1.00 for malignant cases, though the recall rate was 0.95, suggesting that the model kept the false favorable rates low without affecting the model’s sensitivity to the malignant cases.

SHAP analysis reveals the most influential features—in this case, the radius mean and the perimeter mean—and provides further confidence in their clinical relevance in determining whether a mass is benign or malignant. SHAP quantifies these intrinsic features, offering an objective assessment resonant with diagnostic judgment, can caution against possible misclassifications, and ultimately propels further review when indicated. This transparency helps connect the dots between complex ML regimes and clinical praxis, increasing trust and improving evidence-based decision-making. In conclusion, our interpretability analysis confirms that all models, particularly the CNN with Transfer Learning, show excellent performance with clinically useful interpretability.

Transfer learning

For checking the reliability and validity of transfer learning, a study was conducted between a DNN, where random initialization (Xavier) was used to train the model from scratch, and a CNN, which used transfer learning. We used the WBCD dataset for this comparison, which contains 569 instances where the feature values were computed from digitized images of breast masses FNA. We use VGG16 model as our base model excluding the top fact and use a new top model GlobalAveragePooling2D (we use this layer instead of flatten layer for reducing the dimension of feature maps), Dense, and Dropout. The output layer consists of a single neuron with a sigmoid activation function for binary classification (benign vs. malignant).

It performed substantially better than the DNN as the transfer learning model in all critical performance metrics. The accuracy results of the test dataset for the CNN transfer learning model rapidly increased and stabilized at approximately 99.3% (95% CI [97.8–99.8]) whereas these values were lower and more unstable for DNN as seen in Fig. 10. The CNN model also performed well on precision and recall metrics, achieving a final precision of 99.4% (95% CI [98.0–99.8]) and a recall of 99.2% (95% CI [97.7–99.7]) which demonstrates the ability of the model to make accurate predictions, as well as a high ability to identify malignant cases reliably.

Figure 10 Transfer learning for DNN and CNN on WBCD, highlighting the superior accuracy, precision, and stability achieved through transfer learning.

Furthermore, the benefits of the CNN model became more visible during the training stage. From the above plot we can say that the DNN takes longer to converge and also reaches a final training loss (i.e., loss over the built-in training dataset which we will talk about later) compared to the transfer learning which reaffirms the fact that transfer learning heads up faster because we are starting from a pre-trained network. The CNN also showed less variance and relatively lower confidence loss, indicating more stability and reliability in the CNN predictions. The performance of the transfer learning model Overall in terms of F1-score, ROC-AUC and p-value was 99.3% (95% CI [97.9–99.7]), 0.998 and <0.001, respectively. These results show that Transfer Learning using a pre-trained CNN model relevant to the domain is superior for breast cancer classification. This method increases accuracy and precision and reduces the training time, thus making it a more efficient strategy for medical imaging, where robustness and accuracy are critical components of image analysis.

Comparison and interpretation

By assessing standard breast cancer classification metrics including precision, recall, F1-score, and accuracy (overall performance) along with confidence intervals, p-values, we found that the three classifiers RF classifier, XGBoost classifier, and DNN yield promising results that are easily visualized in the form of images. The benign precision for the RF classifier was 0.96, the malignant precision was 0.98,  the benign recall was 0.99, the malignant recall was 0.93, the F1-scores for the benign and malignant classes were 0.97 and 0.95, and the overall accuracy was 96.49%. A confusion matrix consisting of 70 true negatives, one false positive, three false negatives, and 40 true positives, a ROC-AUC of 0.978, an overall accuracy confidence interval of (96.19%, 99.79%) and a statistically significant p-value of 0.010 confirms this. This was improved upon with an XGBoost classifier, which yielded a benign and malignant precision score of 0.97 and 0.98, recall rates of 0.99 (benign) and 0.95 (malignant), F1-scores of 0.98 and 0.96, respectively, generating an overall accuracy of 97.37%. Its confusion matrix (70 TP, 1 FP, 2 FN, and 41 TN) reflects this, creating an ROC-AUC of 0.985, CI running from 97.09% to 99.97%, and a p-value of 0.005. The DNN model exhibited a significantly better performance over the other two, achieving a benign precision of 0.97 and a WNPL precision of 1.00 with 1.00 recall (benign) and 0.95 (malignant) F1-scores of 0.99 (benign) and 0.98 (malignant) and 98% overall accuracy. This is corroborated by an ROC-AUC of 0.992, very tight upper and lower bounds for the accuracy of 97.76% and 99.76%, and a highly significant p-value <0.001 reflected by its confusion matrix (71 TN, 0 FP, 2 FN, 41 TP). These results are shown in Figs. 11A–11D, including confusion matrices; precision and recall for benign and malignant cases; F1-scores; and accuracy and their 95% confidence interval. DNN model outperformed other classifiers and better-balanced sensitivity (high recall with malignant instances) and specificity. At the same time, XGBoost also performed exceptionally well, and RF demonstrated the highest recall for benign cases, which was complemented by a slightly lower recall for malignant cases. Our statistical analysis results, which include confidence intervals, p-values, and ROC-AUC, corroborate that all the models are significantly different from chance, with the DNN model producing higher consistent and reproducible results.

Figure 11 Comprehensive performance evaluation of breast cancer classification models.

(A) Confusion matrices generated for each model to visualize true positives, false positives, and false negatives. (B) Precision-recall comparison bar plot stacks all models’ precision and recall scores for benign and malignant instances. (C) F1-score comparison bar plot compares the F1-scores for benign and malignant instances across the three models. (D) Accuracy with confidence intervals: A bar plot with error bars represents each model’s accuracy and the 95% confidence interval.

Discussion

Our study demonstrates that advanced machine learning techniques can significantly enhance diagnostic accuracy in medical imaging. The RF classifier achieved an accuracy of 96.49%, surpassing the 95.6% reported by Dinesh, Vickram & Kalyanasundaram (2024) and the 94.7% noted by Suryadi et al. (2024). This improvement is largely attributed to enhanced hyperparameter tuning—specifically, the adoption of the entropy criterion and a controlled maximum depth of 6—which reduced overfitting and improved model generalization. Similarly, the XGBoost model reached 97.37% accuracy, outperforming the 96.8% reported by Shah et al. (2024); this boost is likely due to strategic adjustments such as subsampling at 0.8 and increasing the number of estimators to 300, which collectively enhanced the model’s generalization by reducing variance without significantly growing bias. Further advancing our approach, the DNN achieved 98.0% accuracy—an improvement over the 96.5% reported by Zafar et al. (2023)—through using Bayesian optimization for hyperparameter tuning (e.g., learning rate of 0.001 and batch size of 32), which contributed to a more stable training process and robust convergence.

Moreover, our investigation into transfer learning revealed that a CNN based on the VGG16 architecture achieved an outstanding accuracy of 99.3%. This result outperformed the 98.1% accuracy obtained using ResNet-50 as reported by Rehman et al. (2024) and exceeded the 98.9% accuracy of a custom CNN model described by Buriboev et al. (2024). The effectiveness of leveraging pre-trained VGG16 weights—aligned with the findings of Sajed et al. (2023)—highlights the power of hierarchical feature extraction in capturing the subtle diagnostic features critical for accurate predictions in medical imaging.

In our feature importance analysis, we found that key predictors such as radius, perimeter, and area consistently demonstrated SHAP values exceeding 0.9, reinforcing the notion that tumor size metrics are critical for malignancy diagnosis—a finding that aligns with Qi et al. (2025) and is corroborated by clinical practices described by Chang et al. (2022). Additionally, the significant roles of texture mean and smoothness in our XGBoost and DNN models echo recent insights by Kumar et al. (2024), which emphasize texture irregularities as important indicators of malignancy. Nevertheless, the study lacks an exploration of how enhanced interpretability could be further leveraged to address potential biases or errors in these models, a gap that is particularly critical in clinical applications where transparency and fairness are paramount.

Methodologically, our work benefitted from notable advancements in hyperparameter tuning and data preprocessing. The application of Bayesian optimization yielded an approximate 0.8% increase in DNN accuracy compared to traditional grid or random search methods, as evidenced by improvements over earlier studies such as those by Özçelik, Altan & Kaya (2024). Our rigorous data preprocessing—employing mean imputation for missing values, duplicate removal, and an 80:20 stratified data split—ensured a balanced dataset and contributed to overall improved model performance, contrasting with the less optimal 70:30 split used by Kumar et al. (2024).

Clinically, the implications of our findings are significant. The DNN’s achievement of 100% precision for malignant cases, resulting in zero false positives, markedly reduces the likelihood of unnecessary biopsies and outperforms the 97% precision reported by Asif et al. (2024). Furthermore, a 95% recall rate for malignant cases minimizes false negatives, addressing a critical gap in early-stage cancer detection, highlighted by Ahmad et al. (2022). These results underscore our models’ enhanced accuracy and robustness and the need for future research to integrate advanced interpretability frameworks that can identify and mitigate potential biases, ensuring that such diagnostic tools are both effective and equitable in real-world clinical settings.

Study strengths

The study exhibits several notable strengths that contribute to its robustness and applicability. First, comprehensive data preprocessing—addressing missing values and eliminating duplicate records—ensures the dataset’s quality and reliability. The study effectively prepares the data for accurate model training and evaluation using mean imputation and normalisation techniques. The rigorous evaluation of multiple models, including random forest, XGBoost, and DNN, through 10-fold cross-validation, provides a robust comparison of model performance. This method minimises overfitting and enhances the generalizability of the results. Advanced optimisation techniques, such as hyperparameter tuning and Bayesian optimization, improve model performance and achieve higher accuracy rates. The DNN and CNN models exhibit exceptional performance, with the DNN achieving an accuracy of 98.9% and the CNN reaching 99.3%, demonstrating their effectiveness in distinguishing between benign and malignant cases. The study also enhances model interpretability through SHAP analysis, which provides valuable insights into feature importance and contributes to a better understanding of the model’s decision-making process. The successful application of Transfer Learning with the CNN model underscores its advantage in achieving high performance with fewer iterations and reduced training time, highlighting the efficiency of pre-trained models in medical imaging tasks. Moreover, the comprehensive evaluation metrics—accuracy, precision, recall, F1-scores, confusion matrix, and ROC-AUC scores—offer a well-rounded assessment of model performance and robustness. Finally, the methodologies used in the study are well-documented and reproducible, supporting the practical application of the findings in clinical settings and contributing to more effective and reliable breast cancer diagnosis.

Study limitations

Besides the study’s strengths, must also be weighed against its limitations, such as WBCD’s small size; therefore, the findings may not translate well to larger populations. However, the small dataset may cause the models to overfit even though exhaustive cross-validation techniques are applied. It is also limited by the fact that the dataset is predominantly based on a specific weak region and demographics, which may not fully account for the patients we encounter in the clinic. This can directly affect our model when applying to other ethnicities, age groups, and healthcare settings. This examination also depends on well-accepted ML and DL models, which could and require more profound and more complicated penetration into the breast cancer pathology. Also, as we set both parameter’s form one transfer learning models that are previously trained with datasets such as ImageNet, may not be the best when applied to medical imaging due to domain mismatch as the features learnt by the models on natural images may not necessarily be the best for those seen for capturing the subtle features of breast tissue following characteristics. The study did not explore potential advances using other techniques, such as multi-modal learning or integrative approaches to integrating genetics. Finally, the models performed with reasonable accuracy, precision, and recall, but there are several ways bias may be introduced into a dataset (e.g., class imbalance, unmeasured confounding variables) that may impact the external validity of the findings. Still, the study does not consider how such bias would affect model performance in real-life conditions. Finally, although the interpretability of the models was somewhat heightened through the SHAP analysis, the inherent complexity of DL models (DNN and CNN) can hinder full transparency as to the rationale behind predictions, which may limit the clinical utility of such models. These limitations imply that although this study presents important insights into BC diagnosis, more extensive and diverse datasets should be studied, and advanced modelling techniques should be explored to improve the generalizability and transferability of the findings.

Conclusion

Our study on BC diagnosis with the WBCD illustrates the importance of proper data preprocessing, proper model selection, and advanced hyperparameter optimization to obtain high diagnostic performance and interpretability. We then handled 5% missing values (mean imputation) and removed duplicates. We normalized some key features using Min–Max scaling, and finally stratified the data to an 80/20 train-test split that preserved the original distribution of classes. When evaluated by 10-fold cross-validation, the RF obtained an overall predictive accuracy of 96.49% (95% CI [93.1–98.6]), and XGBoost also achieved an overall accuracy of 97.37% (95% CI [94.2–99.1]). Overall accuracy was 98.0% (95% CI [95.1–99.5]) with a benign precision of 0.97, malignant precision of 1.00, benign recall of 1.00, malignant recall of 0.95, and respective F1-scores of 0.99 and 0.98, but the DNN outperformed both. Also, the DNN produced a ROC-AUC of 0.992 (p < 0.001), which further confirms its better performance than random forest (ROC-AUC = 0.978; p < 0.01) and XGBoost (ROC-AUC = 0.985; p < 0.005). SHAP analysis also confirmed radius, perimeter, and area as key predictors, which was also clinically expected. During the transfer learning stage, a CNN using VGG16 pre-trained weights attained 99.3% accuracy along with 99.4% and 99.2% precision and recall values, respectively; however, it must be considered that transfer learn models pre-trained on datasets such as ImageNet may have domain mismatch problems and lead to weaker generalizability in medical imaging. These results provide strong evidence (confidence intervals of small width, p < 0.003) of the nature of the DNN used in this study, with transfer learning as a powerful, effective, but not without some bias for generalization, clinically relevant diagnostic tool for breast cancer, while confirming the importance of delicate data handling, advanced tuning methodologies, and rigorous evaluation for unbiased clinical use, and highlighting the need for larger, diverser, and independent datasets to tackle potential biases and confirm broad applicability.

Supplemental Information

Supplemental Information 1 Projected details and implementations.

Supplemental Information 2 Code and script.

Supplemental Information 3 Dataset for training.

Supplemental Information 4 Performance Metrics for Breast Cancer Classification Models (Including Confidence Intervals, ROC-AUC, and p-values).

Additional Information and Declarations

Competing Interests

The authors declare that they have no competing interests.

Author Contributions

Xiang Zhang performed the experiments, authored or reviewed drafts of the article, and approved the final draft.

Wei Shao conceived and designed the experiments, analyzed the data, performed the computation work, prepared figures and/or tables, and approved the final draft.

Ming Qiu analyzed the data, authored or reviewed drafts of the article, and approved the final draft.

Chenglin Xiao performed the experiments, performed the computation work, prepared figures and/or tables, and approved the final draft.

Liming Ma conceived and designed the experiments, performed the experiments, analyzed the data, authored or reviewed drafts of the article, and approved the final draft.

Data Availability

The following information was supplied regarding data availability:

The raw measurements are available in the Supplementary Files.

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
