# Peer review of "Advanced deep learning and transfer learning approaches for breast cancer classification using advanced multi-line classifiers and datasets with model optimization and interpretability"

_PeerJ Computer Science, doi:10.7717/peerj-cs.2951_

## Round 0.1 · original submission · Major Revisions

While the manuscript explores a relevant area of breast cancer diagnosis using deep learning and transfer learning, it suffers from several critical weaknesses that necessitate major revisions. The primary concerns revolve around dataset limitations, lack of clear novelty and justification for the chosen methods, insufficient validation, and weak presentation of results and discussion. The combination of these issues significantly hinders the manuscript's potential impact.
Addressing all reviewer comments is crucial for the manuscript to be reconsidered for publication. The limited dataset and the lack of a clearly defined novel contribution are the most pressing issues.

Reviewer 1 ·

Basic reporting

The manuscript entitled “Advanced deep learning and transfer learning approaches for breast cancer classification using advanced multi-line classifiers and datasets with model optimization and interpretability” explores the application of deep learning, machine learning, and transfer learning techniques for breast cancer diagnosis. By evaluating several classifiers, including Random Forest, XGBoost, and DNN, and integrating SHAP for model interpretability, the study seeks to address critical issues in accuracy and clinical applicability. While the manuscript demonstrates potential, significant deficiencies in methodology, dataset generalizability, and presentation hinder its impact.
1) The Wisconsin Breast Cancer Dataset (WBCD) used is relatively small, consisting of only 569 instances. This raises concerns about overfitting and the generalizability of the models to diverse populations.
2) The dataset lacks demographic diversity, as it predominantly represents cases from a specific geographic region. This limitation undermines the applicability of the findings to broader, real-world clinical settings.
3) While the study effectively compares multiple classifiers, it fails to articulate a clear and compelling novelty. The use of SHAP for interpretability and transfer learning is well-documented in similar studies.
4) The manuscript does not adequately address how the proposed methods advance the state of the art beyond incremental improvements in accuracy.

Experimental design

5) The validation relies heavily on 10-fold cross-validation, which, while useful, does not compensate for the lack of external validation datasets. Additional validation on independent or real-world datasets would strengthen the study's claims.
6) The evaluation metrics are comprehensive (accuracy, precision, recall, F1-score), but statistical significance tests (e.g., confidence intervals, p-values) are missing, which weakens the robustness of the findings.
7) The transfer learning approach involving a pre-trained CNN is presented as a key innovation, but the specific advantages over the DNN model are not sufficiently explored or justified.
8) The manuscript does not discuss the potential limitations of transfer learning, such as domain mismatch between pre-trained models (e.g., ImageNet) and the target dataset.
9) While SHAP is used to enhance interpretability, the practical implications of interpretability in clinical decision-making are not discussed in depth.

Validity of the findings

10) The study lacks an exploration of how interpretability can address potential biases or errors in the models, especially for critical clinical applications.
11) “Discussion” section should be added in a more highlighting, argumentative way. The author should analysis the reason why the tested results is achieved.
12) The authors should clearly emphasize the contribution of the study. Please note that the up-to-date of references will contribute to the up-to-date of your manuscript. The studies named- “Overcoming nonlinear dynamics in diabetic retinopathy classification: A robust AI-based model with chaotic swarm intelligence optimization and recurrent long short-term memory; Machine learning approach for early diagnosis of Alzheimer's disease using rs-fMRI and metaheuristic optimization with functional connectivity matrices”- can be used to explain the methodology in the study or to indicate the contribution in the “Introduction” section.
13) Figures, such as confusion matrices and SHAP visualizations, are inadequately annotated. They require clearer labels, captions, and integration with the main text to enhance their explanatory value.
14) The presentation of results in tables is sufficient but lacks commentary on key insights or trends, particularly concerning comparisons across models.

Reviewer 2 ·

Basic reporting

1. What challenges did the researchers encounter when applying the CNN models with different layers, and how were these addressed?
2. The contribution lacks clarity in its articulation.
3. Your contribution is not clear please Highlight your contribution in details. You mentioned some general steps; you should explain your method or your algorithms and what you developed or enhanced because there are many topics in CAD. Please explain it.

Experimental design

1. add flow chat for your work.

Validity of the findings

1. Mathematical infrastructure should be strengthened. In particular, consistency between equations must be ensured.
2. Kindly give all the details of datasets and put some examples of the data.

Additional comments

1. What are the Strengths point and Weaknesses point for each case of the algorithm.

---

## Round 0.2 · accepted · Accept

Dear authors, we are pleased to verify that you meet the reviewer's valuable feedback to improve your research.

Thank you for considering PeerJ Computer Science and submitting your work.

Kind regards
PCoelho

Reviewer 1 ·

Basic reporting

All my comments have been thoroughly addressed. It is acceptable in the present form.

Experimental design

All my comments have been thoroughly addressed. It is acceptable in the present form.

Validity of the findings

All my comments have been thoroughly addressed. It is acceptable in the present form.